# The Association between Patient’s Age and Head and Neck Cancer Treatment Decision—A Population-Based Diagnoses-Related Group-Based Nationwide Study in Germany

**DOI:** 10.3390/cancers15061780

**Published:** 2023-03-15

**Authors:** Mussab Kouka, Isabel Hermanns, Peter Schlattmann, Orlando Guntinas-Lichius

**Affiliations:** 1Department of Otorhinolaryngology, Jena University Hospital, 07747 Jena, Germany; 2Department of Medical Statistics, Computer Sciences and Data Sciences, Jena University Hospital, 07743 Jena, Germany

**Keywords:** age, diagnoses-related-group-based, retrospective analysis, treatment rate, head and neck cancer

## Abstract

**Simple Summary:**

The impact of the patient’s age on head and neck cancer (HNC) treatment decision has rarely been studied worldwide. Older age cohorts account for a high proportion of HNC patients and might influence the inpatient treatment decision (surgery, radiotherapy, chemotherapy/biologicals). This population-based study in Germany between 2005 and 2018 included data of 1,226,357 HNC cases. Negative binomial regression was performed to study the time-trend analysis on treatment decision. Older age cohorts (≥65 to <80 years and ≥80 years) predominantly have led to an increase in the treatment rates (biopsy, surgery, radiotherapy, chemotherapy/biologicals), younger ones (≥35 to <50 years and ≥50 to <65 years) to a decrease.

**Abstract:**

Investigations on the association between patient’s age and head and neck cancer (HNC) treatment decision are sparse. Nationwide diagnoses-related group-based data of 1,226,357 cases hospitalized with primary HNC in Germany from 2005 to 2018 were included. Negative binomial regression was performed to study the development of the treatment rates over time. For all treatment options, i.e., biopsies, surgery, radiotherapy, and chemotherapy/biologicals, increases in the treatment rates were seen in patients >80 years (surgery: oral cavity: relative risk [RR]: 1.2, CI: 1.13–1.20; oropharynx: RR: 1.2, CI: 1.15–1.34; hypopharynx: RR: 1.1, CI: 1.02–1.17; larynx: RR: 1.1, CI: 1.04–1.12; radiotherapy: oral cavity: RR: 1.1, CI: 1.07–1.23; oropharynx: RR: 1.3, CI: 1.16–1.49; hypopharynx: RR: 1.3, CI: 1.21–1.46; larynx: RR 1.2, CI: 1.03–1.29; chemotherapy: oral cavity: RR: 1.2, CI: 1.06–1.31; salivary glands: RR: 1.3, CI: 1.09–1.50; oropharynx: RR: 1.4, CI: 1.12–1.83; hypopharynx: RR: 1.3, CI: 1.06–1.48; larynx: RR: 1.3, CI: 1.08–1.52, all *p* < 0.05). Older age cohorts (≥80 years) need more awareness as they are mainly responsible for the increase in the rates of surgery, radiotherapy, and chemotherapy/biologics in HNC patients.

## 1. Introduction

Patients with head and neck cancer (HNC) are nearly 60% already over 60 years old. Of these, approximately 70% are over 65 years and 50% over 70 years old [1,2]. This is important because most clinical trials, especially phase III trial, on different treatment concepts (surgery, radiotherapy, chemotherapy, biologicals, and combinations) typically exclude older patients [3,4]. Hence, older patients are underrepresented in clinical trials [5,6]. The trials build the bases for clinical guidelines on treatment selection but cannot give high-evidence recommendations for older patients. This results in a feeling of insecurity and might explain the reported variability of treatment decision in older patients. However, the global cancer burden of HNC is increasing due to population aging and population growth [7]. Despite of this, clinicaltrials.gov only lists at the moment eleven trials focused on elderly HNC patients, and only one of the studies is a phase III trial (https://www.clinicaltrials.gov/ct2/results?cond=head+and+neck+cancer&term=elderly+old&cntry=&state=&city=&dist=, accessed on 22 February 2023). Very few studies have examined the impact of age on HNC treatment decision and changes over time using large register data [8]. Moreover, age at HNC diagnosis is an important issue as it directly affects survival rates independent of the chosen treatment concept [9,10,11]. Epidemiological trends of HNC with a focus on age regarding gender disparities were already investigated [12]. The influence of age on HNC treatment decision remains under-explored.

In Germany, most inpatient treatments are classified by the German diagnosis-related groups (DRG) system for reimbursement of hospital costs by the health insurance companies. All diagnoses are recorded in the DRG statistics according to *International Classification of Diseases for Oncology*, 10th revision, German modification (ICD-10-GM) of inpatients in Germany. Operations and other medical procedures are also listed via the operation and procedure code (OPS, version 2005 to 2018). With the DRG data, an overview is available that covers almost all hospital treatments (except for psychiatric facilities and hospitals of the German armed forces) in Germany. Recently, we used DRG data from the years 2005 to 2018 to analyze in general treatment trends and mortality during in-hospital treatment for HNC in Germany [13,14]. The aim of the current study was to investigate the influence of patient’s age on inpatient treatment decision of HNC using the same powerful source of DRG data. The results of the study can provide a basis for better understanding if and why age impacts decisions for optimal HNC treatment.

## 2. Methods

### 2.1. Ethical Considerations

There was no need of approval of a local ethics committee. Data of the German Federal Bureau of Statistics (DESTATIS) were used. The anonymized DRG statistics can be accessed via controlled remote data processing of the Research Data Centre (FDZ) of the Federal Statistical Office. The FDZ specifies precise regulations for the provision of the data. This ensures that the requirements of the Federal Statistics Act (BStatG) on anonymization of statistical data are implemented. No results are published that allow conclusions to be drawn about individual cases.

### 2.2. Patients

All patients who were hospitalized with primary HNC in Germany from January 2005 to December 2018 (period of 14 years) were included. A total of 1,226,357 HNC cases were registered. The cases were divided according to the ICD-10-GM into C01–C06 (oral cavity), C07–C09 (salivary glands), C10 (oropharynx), C11 (nasopharynx), C12–C13 (hypopharynx), and C32 (larynx). The DRG statistics provided a data record for each treatment case in which a principal diagnosis according to ICD can be found. In addition, the dataset provided information about gender and age. The frequency of different treatments (biopsy, surgery of the primary tumor, neck dissection, radiotherapy and chemotherapy/biological therapy) per HNC entity in the different age cohorts (≥35 to <50 years; ≥50 to <65 years; ≥65 to <80 years; >80 years) was investigated. Biopsies as tumor biopsies taken during panendoscopy were also included, although taking biopsies is not a treatment in the strict sense. Panendoscopy including taking biopsies is a standard upstream diagnostic procedure in general anesthesia for treatment decision making in Germany. Therefore, the procedures of taking these biopsies were also analyzed in this study. The age cohort < 35 years was excluded due to insufficient case numbers and thus compromising confidentiality status. The different ICD codes were linked to OPS codes: 1–41, 1–42, 1–43, 1–53, and 1–54 = biopsy; 5–21, 5–22, 5–25, to 5–31 = surgery of the primary; 5–401 and 5–403 = neck dissection; 8–52 = radiotherapy and 8–54 = chemotherapy/biological therapy. In addition, the entities of HNC with the highest number of cases were analyzed according to the treatment rates over time (according to the oral cavity, salivary glands, nasopharynx, oropharynx, hypopharynx, and larynx).

### 2.3. Statistical Analysis

Descriptive statistics were performed using SPSS Statistics version 25 (IBM Deutschland GmbH, 71139 Ehningen, Germany). Treatment rates were calculated according to the following arithmetic operation: Cases × 100,000/population and represents the basis for evaluation. Average treatments were calculated for each of the treatment rates from 2005 to 2018. To assess the change in treatment rates over time, negative binomial regression with the respective cases as dependent variable and natural logarithm of the population as an offset was used. Time from 2005–2018 on a yearly basis was used to investigate the time trend. Furthermore, entities and age cohorts were considered. Based on this model, the standard error (StdErr), *p*-value, relative risk (RR), and upper and lower 95% confidence intervals (CI; RR_lower and RR_upper) were estimated and reported. The RR was calculated relative to the years 2005 to 2018. In each case, the RR shows the change in treatment rate over the 14-year period from 2005 to 2018. The significance level of *p* = 0.05 was set. These calculations were performed with SAS 9.4 proc genmod (SAS Institute Inc., Cary, NC, USA).

## 3. Results

In total, 1,226,357 cases of patients with HNC in German hospitals between 2005 and 2018 were examined: 217,859 for biopsy, 378,090 for surgery of the primary, 151,636 for neck dissection, 237,728 for radiotherapy, and 241,044 cases for chemotherapy/biological therapy were included. Table 1 shows the average treatment rates per 100,000 population of the different age cohorts for HNC treatment from 2005 to 2018. Annual treatment rates over time from 2005 to 2018 according to the different age cohorts are shown in Figure 1. HNC patients treated with surgery of the primary in the age cohorts ≥50 to <65 years and ≥65 to <80 years had the highest average treatment rate (≥50 to <65 years: 12.16 ± 8.12; ≥65 to <80: 13.50 ± 10.54). Overall, the neck dissection, radiotherapy, and chemotherapy rates were much lower than the surgery rates. The treatment rates for the youngest age cohort (≥35 to <50 years) were the lowest for all treatment types. In particular, the radiotherapy and chemotherapy rates were dominated by the age cohorts ≥50 to <65 years and ≥65 to <80.

### 3.1. Influence of Age on Biopsy Rates for HNC

The impact over time, the associated regression analyses, and RR including the different age cohorts for biopsy rates for HNC are shown in Table 2. For all entities, the rates of biopsy increased over time. Salivary gland cancer had the most significant increase in treatment rate (RR: 1.35, CI: 1.29–1.4, *p* < 0.0001). For all entities, the older age cohorts ≥65 to <80 years and ≥80 years contributed to the positive trend (all *p* < 0.01). The youngest age cohort (≥35 to <50 years) resulted in a decrease for HNC of the oral cavity, oropharynx, hypopharynx, and larynx (oral cavity: RR: 0.9, CI: 0.86–0.97, *p* = 0.0030; oropharynx: RR: 0.85, CI: 0.80–0.91, *p* < 0.0001; hypopharynx: RR: 0.7, CI: 0.68–0.81, *p* < 0.0001; larynx: RR: 0.9, CI: 0.82–0.92, *p* < 0.0001). Overall, increases in the treatment rates were seen for all tumor entities for the cohorts ≥50 to <65 years, ≥65 to <80 years, and >80 years except for the hypopharynx in the cohort ≥50 to <65 years (*p* < 0.05). The age cohort >80 years had the most significant increase in the treatment rates among all age cohorts for all entities except for the salivary glands (oral cavity: RR: 1.3, CI: 1.29–1.41, *p* < 0.0001; oropharynx: RR: 1.43, CI: 1.32–1.55, *p* < 0.0001; nasopharynx: RR: 1.22, CI: 1.06–1.42, *p* = 0.006; hypopharynx: RR: 1.32, CI: 1.23–1.42, *p* < 0.0001; larynx: RR: 1.35, CI: 1.29–1.43, *p* < 0.0001). The treatment rates decreased only in the age cohort ≥35 to <50 years for all tumor entities except for the salivary glands and nasopharynx (all *p* < 0.01).

### 3.2. Influence of Age on Surgery of the Primary Rates for HNC

The impact over time, the associated regression analyses, and the RR including the different age cohorts for surgery of the primary rates for HNC are shown in Table 3. The most significant change over time was seen in patients with hypopharynx cancer. Here, a decrease in surgery rates was seen (RR: 0.91, CI: 0.88–0.94, *p* < 0.0001). The localizations of the oral cavity and salivary glands showed increasing surgery rates (oral cavity: RR: 1.1, CI: 1.03–1.10, *p* = 0.0001; salivary glands: RR: 1.1, CI: 1.04–1.10, *p* < 0.0001). In contrast, the nasopharynx and larynx showed decreasing surgery rates (nasopharynx: RR: 0.92, CI: 0.88–0.95, *p* < 0.0001; larynx: RR: 0.96, CI: 0.94–0.98, *p* = 0.0004). The age cohorts ≥35 to <50 years, ≥50 to <65 years, and ≥65 to <80 years resulted in a decrease. HNC of the oropharynx did not have a significant change (*p* = 0.7633). Overall, the increases in the treatment rates were seen for all tumor entities for the cohorts ≥65 to <80 years and >80 years except for the nasopharynx in the cohort ≥65 to <80 years (all *p* < 0.05). The age cohort >80 years had the most significant increase in the treatment rates among all age cohorts for all entities except for the salivary glands and nasopharynx (oral cavity: RR: 1.2, CI: 1.13–1.20, *p* < 0.0001; oropharynx: RR: 1.2, CI: 1.15–1.34, *p* < 0.0001; hypopharynx: RR: 1.1, CI: 1.02–1.17, *p* = 0.0130; larynx: RR: 1.1, CI: 1.04–1.12, *p* < 0.0001). The treatment rates decreased in the age cohort ≥35 to <50 and ≥50 to <65 years for all tumor entities except for salivary glands (all *p* < 0.05).

### 3.3. Influence of Age on Neck Dissection Rates for HNC

Table 4 shows the corresponding statistical analysis with influence over the time and age cohorts for the neck dissection rates. Oral cavity and salivary gland cancer showed a positive trend in the treatment rates (oral cavity: RR: 1.1, CI: 1.06–1.11, *p* < 0.0001; salivary glands: RR 1.1, CI: 1.03–1.10, *p* = 0.0005). The most significant change over time was in HNC of hypopharynx (RR: 0.9, CI: 0.82–0.90, *p* < 0.0001). The age cohort ≥35 to <50 years had a particular impact among all cohorts on HNC of hypopharynx (RR: 0.6, CI: 0.59–0.71, *p* < 0.0001). HNC of the oropharynx, nasopharynx, and hypopharynx consistently showed decreasing treatment rates (oropharynx: RR: 1.0, CI: 0.93–0.99, *p* = 0.0096; nasopharynx: RR 0.6, CI: 0.42–0.74, *p* < 0.0001; hypopharynx: RR: 0.9, CI: 0.82–0.90, *p* < 0.0001). The age cohorts ≥35 to <50 years, ≥50 to <65 years, and ≥65 to <80 years had the most impact for decreased treatment rates of HNC of the nasopharynx and hypopharynx. For HNC of the oropharynx, the age cohorts ≥35 to <50 years and ≥80 years had a significant impact on the negative treatment rates for neck dissections. Overall, the oldest age cohort (>80 years) showed a significant increase in the treatment rates among all age cohorts for HNC of the oral cavity and larynx (oral cavity: RR: 1.3, CI: 1.20–1.31, *p* < 0.0001; larynx: RR 1.1, CI: 1.03–1.21, *p* = 0.007). The treatment rates decreased in the age cohort ≥35 to <50 and ≥50 to <65 years for all entities (all *p* < 0.05).

### 3.4. Influence of Age on Radiotherapy Rates for HNC

Table 5 shows the impact over time, the corresponding regression analyses, and the RR including the different age cohorts for radiotherapy rates for HNC. Overall, there was a slight upward trend for the radiotherapy rates for oral cavity cancer due to the higher age cohorts, especially the age cohort ≥65 to <80 years (RR: 1.2, CI: 1.17–1.22, *p* < 0.0001). The rate of radiotherapy also increased for HNC of the salivary glands and oropharynx (salivary glands: RR: 1.1, CI: 1.05–1.15, *p* < 0.0001; oropharynx: RR 1.1, CI: 1.02–1.14, *p* = 0.0064). HNC of the nasopharynx and hypopharynx had a slight negative treatment trend (nasopharynx: RR: 0.93, CI: 0.88–0.98, *p* = 0.0067; hypopharynx: RR 0.9, CI: 0.88–0.99, *p* = 0.0323). There was no significant change in HNC of larynx (*p* = 0.1779). For HNC of the nasopharynx, the age cohort ≥50 to <65 years was primarily responsible for the negative trend (RR: 0.83, CI: 0.75–0.91, *p* < 0.0001). The other age cohorts resulted in no significant change (all *p* > 0.05). Overall, positive treatment trends were seen for all entities except for the nasopharynx in the older age cohorts ≥65 to <80 years and >80 years (all *p* < 0.05). The oldest age cohort (>80 years) showed a significant increase in the treatment rates among all age cohorts for HNC of the oral cavity, oropharynx, hypopharynx and larynx (oral cavity: RR: 1.1, CI: 1.07–1.23, *p* = 0.0002; oropharynx: RR: 1.3, CI: 1.16–1.49, *p* < 0.0001; hypopharynx: RR: 1.3, CI: 1.21–1.46, *p* < 0.0001; larynx: RR 1.2, CI: 1.03–1.29, *p* = 0.0134). The treatment rates decreased only in the age cohort ≥35 to <50 years for all tumor entities except for the nasopharynx (all *p* < 0.05).

### 3.5. Influence of Age on Chemotherapy/Biological Therapy Rates for HNC

Table 6 contains the corresponding regression analyses including the different age cohorts for the chemotherapy rates for HNC patients. Over the 14 years, there was an increase in the chemotherapy rates for HNC of the oral cavity, salivary glands, and oropharynx (oral cavity: RR: 1.1, CI: 1.05–1.51, *p* = 0.000; salivary glands: RR: 1.1, CI: 1.03–1.18, *p* = 0.0042; oropharynx: RR 1.1, CI: 1.02–1.14, *p* = 0.0113). The treatment rates for hypopharynx cancer showed a slight decreasing trend (RR: 0.93, CI: 0.89–0.98, *p* < 0.0089). The treatment rates decreased in the age cohort ≥35 to <50 years for all tumor entities except for the nasopharynx (oral cavity: RR: 0.8, CI: 0.73–0.88, *p* < 0.0001; salivary glands: RR: 0.9, CI: 0.80–0.96, *p* = 0.0066; oropharynx: RR 0.8, CI: 0.71–0.87, *p* < 0.0001; hypopharynx: RR: 0.6, CI: 0.55–0.70, *p* < 0.0001; larynx: RR: 0.83, CI: 0.77–0.90, *p* < 0,0001). The oldest age cohort (>80 years) was associated with a positive treatment trend for all entities (oral cavity: RR: 1.2, CI: 1.06–1.31, *p* = 0.003; salivary glands: RR: 1.3, CI: 1.09–1.50, *p* = 0.0022; oropharynx: RR: 1.4, CI: 1.12–1.83, *p* = 0.0041; nasopharynx: RR: 1.1, CI: 0.72–1.56, *p* = 0.777; hypopharynx: RR: 1.3, CI: 1.06–1.48, *p* < 0.0087; larynx: RR: 1.3, CI: 1.08–1.52, *p* = 0.0042). Overall, increases in the treatment rates were seen for the oral cavity, salivary glands, oropharynx, hypopharynx, and larynx in the older age cohorts ≥65 to <80 years and >80 years (all *p* < 0.05).

## 4. Discussion

The association between HNC patient’s age on treatment decision over time based on population-based real word registry data has been rarely investigated worldwide. To our knowledge, this study provides for the first time a diagnoses-related-group-based nationwide study investigating the influence of age on treatment rates of HNC patients in Germany. HNC patients treated with surgery of the primary in the age cohorts ≥50 to <65 years and ≥65 to <80 had the highest average treatment rate. Overall, the older age cohorts (≥65 to <80 years and ≥80 years) have seen an increase in the treatment rates (biopsy, surgery, radiotherapy, chemotherapy/biologicals). This increase in elderly patients was most pronounced for biopsies and surgeries. The increase for biopsies can easily be explained by the increasing incidence rates of HNC in elderly patients, thus needing this standard diagnostic procedure. The surgery rates might have increased because screening tools were developed in the recent years, allowing a better risk stratification and individual treatment deintensification decisions in elderly patients [15]. In contrast, a decrease in the treatment rates in younger patients (≥35 to <50 years and ≥50 to <65 years) was seen. We can only speculate about the reasons. This might be a sign of the worldwide reduced tobacco and alcohol consumption. This effect now reaches the younger age cohorts first [16].

Older HNC patients such as other cancer patients have a high comorbidity [17,18,19]. Comorbidity and older age are important risk factors for deviation from guideline-based treatment decision [20]. Older patients are generally treated less aggressively, for instance, with aggressive radiochemotherapy [18,21,22,23,24,25], and older patients and their families are more likely to refuse invasive treatment [26,27]. The DRG dataset used did not allow an analysis of the comorbidity of the patients. Therefore, the shown trends in treatment do not allow an analysis on the impact of comorbidity and other influencing factors, especially in the older HNC patients, on the treatment decision. Our results showed an overall lower treatment rate of surgery and chemotherapy of the oldest age cohort (≥80 years) compared to younger age groups. This might be interpreted as a less aggressive treatment in the oldest age group in Germany. Nevertheless, the relative increase in chemotherapy/biologicals use over time was largest in patients ≥65 to <80 years. The data structure did not allow a differentiation between chemotherapeutics and biologicals. It is rather plausible that the increased use is related to an increasing use of biologicals than of the chemotherapeutics. Cetuximab was licensed in 2004. Cetuximab is especially used in elderly HNC patients with worse performance status who are not ineligible to receive a platinum-based chemotherapy [28]. Thompson-Harvey et al. addressed from the Surveillance, Epidemiology and End Results (SEER), a nationally representative, population-based cancer database, an increasing incidence of higher-stage HNC in the United States. For both sexes, mean age-adjusted incidence rates increased with older age cohorts. The age cohorts ≥50 to 59 years and ≥60 years showed significant increases in incidence rates from 2004 to 2015 [29]. In contrast, Dittberner et al. reported a peak incidence in the male population in Thuringia at the age of 60 to 64 years from 1996 to 2016. The incidence of the female population increased with increasing age continuously and reached the maximum at ≥85 year [12]. This might also explain the increase in the treatment rates in our study with the increasing incidence of the male and female population in older age, as the increase or decrease in the treatment rates also correlates with the increase in incidence rates in Germany. Additionally, Dittberner et al. described an increase in surgery and in radiochemotherapy in Thuringia from 1996 to 2016. These results explain our data of increasing treatment rates in surgery, radiotherapy, and chemotherapy. Dittberner et al. also reported that a decrease in HNC of the hypopharynx, nasopharynx, and larynx was observed in Thuringia. Our data also showed an overall average decrease in the treatment rates for hypopharyngeal and nasopharyngeal cancer for all treatment types except biopsy. The treatment rates for larynx cancer decreased only for surgery and neck dissection.

A previous U.S. American study based on data of 2688 HNC patients in a phase III trial by Kish et al. investigated the impact of age on treatment decision and outcome in locally advanced HNC [8]. A total of 309 patients (11.5%) of the 2688 HNC patients were ≥70 years old. Kish et al. reported that older age (≥70 years) was associated with lower OS. A lower OS in older patients was also seen in other trials and was related to deviations from a standard treatment [18,19]. DRG data are not linked to survival data. Therefore, the present study could not analyze the impact of treatment decision on OS.

The present study had some other limitations. The retrospective design and data collection did not allow for any traceability of coding. Causal connections, for example between cancer diagnosis and chosen therapy, were only traceable to a limited extent. Tumor stage and the HPV status are not coded in the DRG system. The federal tumor registers (not the DRG system) have registered the HPV status for patients with oropharyngeal cancer since introduction of the 8th edition of the TNM classification in 2017. The present study included all cases until 2018. Hence, an influence of the HPV status on decision making could only be marginal. As in many other countries, we see an increase in oropharyngeal cancer in the recent years. This increase is probably related to an increase in HPV+ oropharyngeal cancer [12]. This might explain why the treatment rates for oropharyngeal cancer have increased in the recent years. Furthermore, we could not differentiate between primary treatment and treatment for tumor recurrence. For instance, we cannot explain why chemotherapy/biological treatment was increasingly used in some age cohorts of the patients with salivary gland cancer. Chemotherapy/biological treatment is no standard for the treatment of salivary gland cancer. Hence, we can only speculate that these patients were not treated according to the current guidelines, were possibly treated in a palliative setting, or possibly in clinical trials. Additionally, the data included all discharged, fully hospitalized patients. Almost all cases of surgery for HNC are performed as inpatient cases. In contrast, for about 10 years there has been an increasing tendency to perform radiotherapy, especially when performed as a single modality, as outpatient treatment. If the patients receive radiochemotherapy, most patients are still treated as inpatients in the weeks they receive chemotherapy. The numbers on the portion of outpatient radiotherapy treatment for HNC in Germany are not published. Nowadays, most radiation oncologists would recommend outpatient radiotherapy even in elderly HNC patients [30]. Day-care or outpatient patients were not included. Hence, the overall number of patients receiving radiotherapy for HNC in Germany is probably higher, especially in the most recent years. In general, older HNC patients, especially with comorbidities, might have a higher probability for inpatient treatment. Elective medical services, for example a desired treatment by a chief physician, were not billed via the DRG system. Accordingly, about 10% of the medical services provided by hospitals was not billed via the DRG system. This must be considered when interpreting the data. A higher final treatment rate must be expected [31]. The majority of patients receive several therapies. However, due to anonymization and the lack of pseudonyms, the data did not allow the assignment of several cases to one patient case. Patients who were treated more frequently as inpatients for the same diagnosis are each coded as a new case. Additionally, data of the older age cohorts (65 to <80 years and ≥80 years) with nasopharyngeal cancer treated with neck dissection were not published in 2009, 2012, and 2015 and thus not included in the calculation, as this would have allowed conclusions to be drawn about individual cases due to the low number of cases. However, the DRG statistics provided a comprehensive and powerful dataset that covers the majority of all inpatient cases in Germany. This provided a unique reliable basis of inpatient healthcare. It also offered the advantage of already collected and digitally available data. The results from our population-based study showed that age has an impact on the treatment rates of HNC and is an independent variable. In conclusion, age is not only a chronological number but a measure of physiological and functional factors of elderly patient cohorts, thereby altering HNC treatment rates. As a result, patients of older age receive reduced or inadequate treatment leading to lower OS in HNC patients [32]. There are only a few studies that have investigated the impact of age on HNC treatment [8,33,34]. For a better understanding of the role of chronological age in HNC treatment rates, clinical trials for the older population stratified for comorbidities are needed to provide adequate evidence.

## 5. Conclusions

This study provides for the first time a diagnoses-related group-based nationwide study investigating the influence of age on the treatment rates of 1,226,357 HNC cases in Germany between 2005 and 2018. In summary, the older age cohorts (≥65 to <80 years and ≥80 years) had an increasing influence on the treatment rates, the younger ones (≥35 to <50 years and ≥50 to <65 years) a decreasing one. For chemotherapy, radiotherapy, and surgery, increases in the treatment rates were seen for HNC in the oral cavity, salivary glands, oropharynx, hypopharynx, and larynx in the older age cohorts (≥65 to <80 years and >80 years). The treatment rates decreased only in the age cohort ≥35 to <50 years for all tumor entities except for the nasopharynx in chemotherapy. The results from our study showed that age, especially older age, has an important impact on HNC treatment rates. Older HNC patients need more awareness in clinical trials. Further studies are needed that include concomitant circumstances, comorbidities, risk factors, and cancer stages in the analyses for a better understanding of the role of chronological age in HNC treatment.

## Figures and Tables

**Figure 1 cancers-15-01780-f001:**
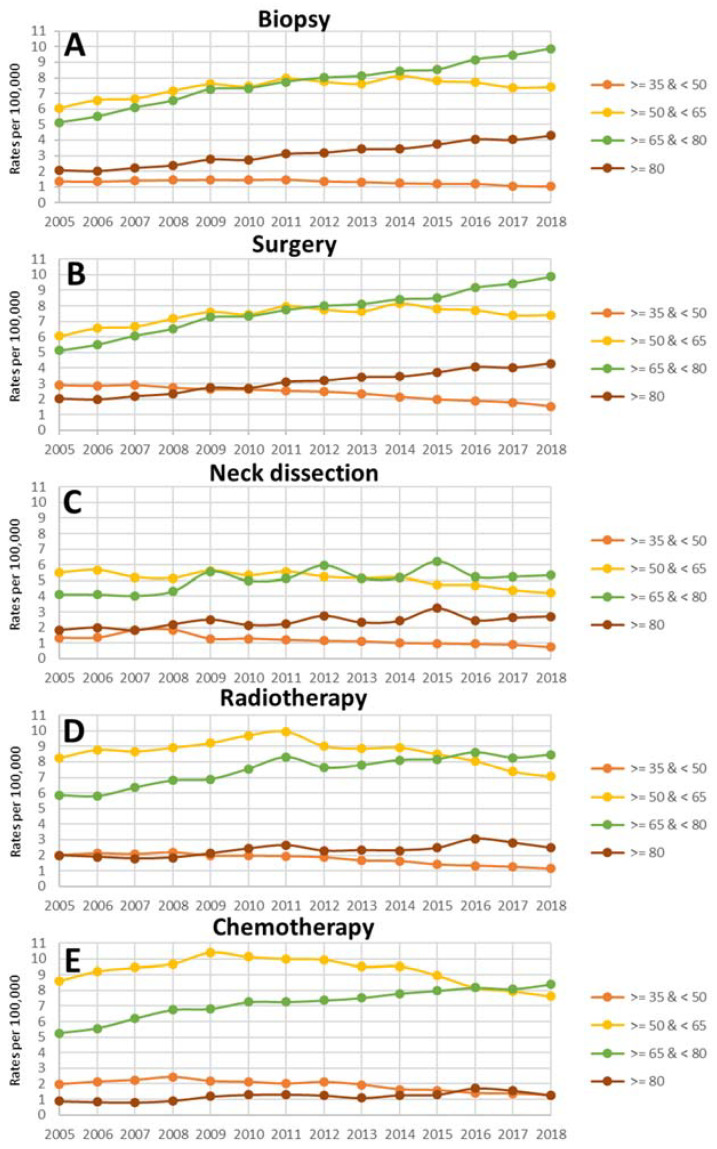
Influence of age in years on treatment rates over time from 2005 to 2018 according to biopsy (**A**), surgery of the primary (**B**), neck dissection (**C**), radiotherapy (**D**) and chemotherapy/biological therapy (**E**).

**Table 1 cancers-15-01780-t001:** Average treatment rates per 100,000 population of different age cohort for HNC treatment in Germany, for the years 2005–2018.

Treatment	Age Cohort (Years)	Mean	SD
Biopsy	all	4.86	4.28
	35–49	1.31	0.62
	50–64	7.38	3.75
	65–79	7.66	4.83
	80+	3.10	2.42
Surgery of the primary	all	8.70	8.52
	35–49	2.39	1.67
	50–64	12.16	8.12
	65–79	13.50	10.54
	80+	6.77	5.83
Neck dissection	all	3.42	3.30
	35–49	1.21	1.03
	50–64	5.12	3.60
	65–79	5.01	3.61
	80+	2.36	2.31
Radiotherapy	all	5.05	4.17
	35–49	1.75	0.87
	50–64	8.65	4.21
	65–79	7.47	3.48
	80+	2.31	1.41
Chemotherapy/biological therapy	all	4.87	4.40
	35–49	1.89	0.90
	50–64	9.23	4.35
	65–79	7.17	3.27
	80+	1.20	0.70

SD = standard deviation.

**Table 2 cancers-15-01780-t002:** Change of biopsy rates for HNC in Germany in relation to the tumor localization and different age cohorts, for the years 2005–2018.

Localization	Age Cohort (Years)	Estimate	StdErr	*p **	RR	(95% CI)
Oral cavity	all	0.0392	0.0029	**<0.0001**	1.21665	(1.18239–1.25191)
	35–49	−0.0183	0.0062	**0.0030**	0.91244	(0.85888–0.96935)
	50–64	0.0221	0.0030	**<0.0001**	1.11657	(1.08423–1.14988)
	65–79	0.0507	0.0024	**<0.0001**	1.28861	(1.25906–1.31887)
	80+	0.0599	0.0043	**<0.0001**	1.34885	(1.29294–1.40718)
Salivary glands	all	0.0597	0.0042	**<0.0001**	1.34812	(1.29362–1.40492)
	35–49	0.0266	0.0057	**<0.0001**	1.14198	(1.08045–1.20701)
	50–64	0.0383	0.0050	**<0.0001**	1.21101	(1.15331–1.27159)
	65–79	0.0740	0.0042	**<0.0001**	1.44774	(1.38986–1.50802)
	80+	0.0588	0.0070	**<0.0001**	1.34151	(1.25220–1.43719)
Oropharynx	all	0.0365	0.0041	**<0.0001**	1.20024	(1.15346–1.24890)
	35–49	−0.0322	0.0068	**<0.0001**	0.85126	(0.79663–0.90965)
	50–64	0.0164	0.0046	**0.0003**	1.08566	(1.03818–1.13531)
	65–79	0.0567	0.0038	**<0.0001**	1.32809	(1.27938–1.37865)
	80+	0.0712	0.0082	**<0.0001**	1.42782	(1.31790–1.54690)
Nasopharynx	all	0.0204	0.0033	**<0.0001**	1.10749	(1.07179–1.14438)
	35–49	0.0213	0.0077	**0.0056**	1.11239	(1.03165–1.19945)
	50–64	0.0056	0.0049	0.2560	1.02844	(0.97986–1.07943)
	65–79	0.0181	0.0079	**0.0226**	1.09454	(1.01278–1.18290)
	80+	0.0408	0.0149	**0.0060**	1.22643	(1.06024–1.41867)
Hypopharynx	all	0.0142	0.0047	**0.0023**	1.07374	(1.02567–1.12406)
	35–49	−0.0597	0.0089	**<0.0001**	0.74190	(0.68019–0.80921)
	50–64	−0.0091	0.0065	0.1588	0.95543	(0.89672–1.01798)
	65–79	0.0378	0.0049	**<0.0001**	1.20825	(1.15177–1.26750)
	80+	0.0558	0.0073	**<0.0001**	1.32213	(1.23027–1.42086)
Larynx	all	0.0307	0.0033	**<0.0001**	1.16604	(1.12869–1.20463)
	35–49	−0.0276	0.0059	**<0.0001**	0.87105	(0.82183–0.92321)
	50–64	0.0094	0.0036	**0.0087**	1.04805	(1.01193–1.08545)
	65–79	0.0357	0.0032	**<0.0001**	1.19552	(1.15915–1.23304)
	80+	0.0607	0.0052	**<0.0001**	1.35484	(1.28814–1.42500)

StdErr = standard error; RR = relative risk; CI = confidence interval; * significant values (*p* < 0.05) in bold.

**Table 3 cancers-15-01780-t003:** Change of surgery of the primary rates for HNC in Germany in relation to the tumor localization and different age cohorts, for the years 2005–2018.

Localization	Age Cohort (Years)	Estimate	StdErr	*p **	RR	(95% CI)
Oral cavity	all	0.0128	0.0033	**0.0001**	1.06623	(1.03194–1.10165)
	35–49	−0.0345	0.0057	**<0.0001**	0.84144	(0.79551–0.89003)
	50–64	−0.0066	0.0036	0.0699	0.96766	(0.93387–1.00268)
	65–79	0.0260	0.0031	**<0.0001**	1.13881	(1.10455–1.17414)
	80+	0.0307	0.0028	**<0.0001**	1.16616	(1.13463–1.19857)
Salivary glands	all	0.0138	0.0027	**<0.0001**	1.07165	(1.04370–1.10036)
	35–49	−0.0063	0.0035	0.0662	0.96876	(0.93652–1.00212)
	50–64	−0.0055	0.0033	0.0967	0.97296	(0.94200–1.00494)
	65–79	0.0204	0.0028	**<0.0001**	1.10722	(1.07702–1.13827)
	80+	0.0102	0.0042	**0.0149**	1.05230	(1.00999–1.09639)
Oropharynx	all	0.0007	0.0025	0.7633	1.00373	(0.97972–1.02831)
	35–49	−0.0663	0.0053	**<0.0001**	0.71781	(0.68164–0.75590)
	50–64	−0.0202	0.0035	**<0.0001**	0.90414	(0.87347–0.93589)
	65–79	0.0235	0.0023	**<0.0001**	1.12470	(1.10000–1.14995)
	80+	0.0428	0.0077	**<0.0001**	1.23848	(1.14837–1.33567)
Nasopharynx	all	−0.0175	0.0036	**<0.0001**	0.91627	(0.88426–0.94944)
	35–49	−0.0268	0.0080	**0.0008**	0.87443	(0.80838–0.94587)
	50–64	−0.0320	0.0055	**<0.0001**	0.85226	(0.80765–0.89932)
	65–79	−0.0189	0.0081	**0.0200**	0.90976	(0.84008–0.98522)
	80+	0.0123	0.0177	0.4867	1.06354	(0.89407–1.26513)
Hypopharynx	all	−0.0193	0.0036	**<0.0001**	0.90781	(0.87649–0.94026)
	35–49	−0.0830	0.0082	**<0.0001**	0.66048	(0.60921–0.71608)
	50–64	−0.0449	0.0056	**<0.0001**	0.79903	(0.75600–0.84450)
	65–79	0.0071	0.0036	**0.0488**	1.03606	(1.00019–1.07321)
	80+	0.0176	0.0071	**0.0130**	1.09173	(1.01867–1.17004)
Larynx	all	−0.0073	0.0021	**0.0004**	0.96411	(0.94484–0.98378)
	35–49	−0.0598	0.0051	**<0.0001**	0.74143	(0.70535–0.77936)
	50–64	−0.0313	0.0017	**<0.0001**	0.85526	(0.84154–0.86921)
	65–79	−0.0000	0.0021	0.9872	0.99983	(0.97975–1.02033)
	80+	0.0149	0.0037	**<0.0001**	1.07748	(1.03934–1.11702)

StdErr = standard error; RR = relative risk; CI = confidence interval; * significant values (*p* < 0.05) in bold.

**Table 4 cancers-15-01780-t004:** Change of neck dissection rates for HNC in Germany in relation to the tumor localization and different age cohorts, for the years 2005–2018.

Localization	Age Cohort (Years)	Estimate	StdErr	*p **	RR	(95% CI)
Oral cavity	all	0.0165	0.0027	**<0.0001**	1.08622	(1.05764–1.11558)
	35–49	−0.0311	0.0059	**<0.0001**	0.85602	(0.80788–0.90702)
	50–64	−0.0041	0.0030	0.1670	0.97963	(0.95146–1.00864)
	65–79	0.0326	0.0028	**<0.0001**	1.17717	(1.14502–1.21022)
	80+	0.0458	0.0044	**<0.0001**	1.25737	(1.20462–1.31243)
Salivary glands	all	0.0119	0.0034	**0.0005**	1.06150	(1.02638–1.09783)
	35–49	−0.0072	0.0055	0.1895	0.96451	(0.91382–1.01801)
	50–64	−0.0108	0.0036	**0.0028**	0.94738	(0.91436–0.98160)
	65–79	0.0224	0.0041	**<0.0001**	1.11839	(1.07406–1.16455)
	80+	0.0082	0.0072	0.2565	1.04171	(0.97072–1.11790)
Oropharynx	all	−0.0077	0.0030	**0.0096**	0.96232	(0.93476–0.99070)
	35–49	−0.1439	0.0297	**<0.0001**	0.48710	(0.36408–0.65168)
	50–64	0.0018	0.0147	0.9029	1.00903	(0.87332–1.16583)
	65–79	0.0473	0.0431	0.2728	1.26687	(0.83013–1.93338)
	80+	−0.0552	0.0313	0.0775	0.75883	(0.55855–1.03093)
Nasopharynx	all	−0.1170	0.0290	**<0.0001**	0.55701	(0.41905–0.74038)
	35–49	−0.1339	0.0305	**<0.0001**	0.51204	(0.37990–0.69015)
	50–64	−0.1476	0.0315	**<0.0001**	0.47803	(0.35107–0.65091)
	65–79	−0.1127	0.0391	**0.0039**	0.56928	(0.38822–0.83479)
	80+	−0.0523	0.0290	0.0717	0.76984	(0.57913–1.02335)
Hypopharynx	all	−0.0301	0.0045	**<0.0001**	0.86049	(0.82331–0.89935)
	35–49	−0.0870	0.0092	**<0.0001**	0.64731	(0.59153–0.70834)
	50–64	−0.0545	0.0055	**<0.0001**	0.76153	(0.72190–0.80335)
	65–79	−0.0014	0.0044	0.7554	0.99321	(0.95151–1.03673)
	80+	0.0155	0.0142	0.2770	1.08036	(0.93981–1.24194)
Larynx	all	−0.0140	0.0054	**0.0099**	0.93239	(0.88407–0.98334)
	35–49	−0.0757	0.0084	**<0.0001**	0.68489	(0.63100–0.74337)
	50–64	−0.0452	0.0022	**<0.0001**	0.79768	(0.78071–0.81501)
	65–79	−0.0060	0.0027	**0.0256**	0.97051	(0.94534–0.99636)
	80+	0.0218	0.0081	**0.0074**	1.11508	(1.02963–1.20762)

StdErr = standard error; RR = relative risk; CI = confidence interval; * significant values (*p* < 0.05) in bold.

**Table 5 cancers-15-01780-t005:** Change of radiotherapy rates for HNC in Germany in relation to the tumor localization and different age cohorts, for the years 2005–2018.

Localization	Age Cohort (Years)	Estimate	StdErr	*p **	RR	(95% CI)
Oral cavity	all	0.0175	0.0039	**<0.0001**	1.09118	(1.04995–1.13402)
	35–49	−0.0422	0.0074	**<0.0001**	0.80992	(0.75353–0.87052)
	50–64	0.0029	0.0053	0.5810	1.01468	(0.96352–1.06855)
	65–79	0.0357	0.0022	**<0.0001**	1.19550	(1.16963–1.22195)
	80+	0.0269	0.0072	**0.0002**	1.14375	(1.06539–1.22787)
Salivary glands	all	0.0190	0.0043	**<0.0001**	1.09962	(1.05445–1.14673)
	35–49	−0.0309	0.0074	**<0.0001**	0.85663	(0.79650–0.92131)
	50–64	−0.0036	0.0052	0.4839	0.98209	(0.93364–1.03305)
	65–79	0.0443	0.0047	**<0.0001**	1.24784	(1.19139–1.30697)
	80+	0.0255	0.0111	**0.0222**	1.13573	(1.01835–1.26664)
Oropharynx	all	0.0156	0.0057	**0.0064**	1.08118	(1.02216–1.14361)
	35–49	−0.0536	0.0074	**<0.0001**	0.76479	(0.71137–0.82222)
	50–64	−0.0060	0.0075	0.4268	0.97064	(0.90184–1.04468)
	65–79	0.0456	0.0045	**<0.0001**	1.25614	(1.20210–1.31260)
	80+	0.0552	0.0126	**<0.0001**	1.31791	(1.16472–1.49124)
Nasopharynx	all	−0.0143	0.0053	**0.0067**	0.93089	(0.88389–0.98039)
	35–49	−0.0013	0.0083	0.8738	0.99345	(0.91600–1.07744)
	50–64	−0.0380	0.0093	**<0.0001**	0.82684	(0.75445–0.90618)
	65–79	−0.0059	0.0097	0.5421	0.97077	(0.88246–1.06791)
	80+	−0.0275	0.0313	0.3799	0.87168	(0.64152–1.18440)
Hypopharynx	all	−0.0133	0.0062	**0.0323**	0.93583	(0.88069–0.99441)
	35–49	−0.0886	0.0102	**<0.0001**	0.64214	(0.58104–0.70967)
	50–64	−0.0363	0.0066	**<0.0001**	0.83394	(0.78151–0.88987)
	65–79	0.0119	0.0061	0.0534	1.06114	(0.99913–1.12701)
	80+	0.0569	0.0094	**<0.0001**	1.32927	(1.21271–1.45703)
Larynx	all	0.0063	0.0047	0.1779	1.03211	(0.98573–1.08067)
	35–49	−0.0529	0.0102	**<0.0001**	0.76755	(0.69436–0.84846)
	50–64	−0.0177	0.0049	**0.0003**	0.91508	(0.87232–0.95995)
	65–79	0.0201	0.0050	**<0.0001**	1.10561	(1.05229–1.16163)
	80+	0.0283	0.0115	**0.0134**	1.15215	(1.02985–1.28899)

StdErr = standard error; RR = relative risk; CI = confidence interval; * significant values (*p* < 0.05) in bold.

**Table 6 cancers-15-01780-t006:** Change of chemotherapy/biological therapy rates for HNC in Germany in relation to the tumor localization and different age cohorts, for the years 2005–2018.

Localization	Age Cohort (Years)	Estimate	StdErr	*p **	RR	(95% CI)
Oral cavity	all	0.0188	0.0049	**0.0001**	1.09840	(1.04732–1.15196)
	35–49	−0.0446	0.0096	**<0.0001**	0.80015	(0.72796–0.87950)
	50–64	0.0036	0.0057	0.5264	1.01836	(0.96261–1.07734)
	65–79	0.0424	0.0045	**<0.0001**	1.23624	(1.18290–1.29198)
	80+	0.0327	0.0110	**0.0030**	1.17761	(1.05694–1.31205)
Salivary glands	all	0.0192	0.0067	**0.0042**	1.10099	(1.03086–1.17588)
	35–49	−0.0267	0.0098	**0.0066**	0.87522	(0.79504–0.96349)
	50–64	0.0001	0.0085	0.9906	1.00050	(0.92063–1.08730)
	65–79	0.0482	0.0043	**<0.0001**	1.27256	(1.22028–1.32707)
	80+	0.0498	0.0162	**0.0022**	1.28248	(1.09378–1.50372)
Oropharynx	all	0.0147	0.0058	**0.0113**	1.07650	(1.01682–1.13968)
	35–49	−0.0480	0.0104	**<0.0001**	0.78674	(0.71034–0.8713)
	50–64	−0.0072	0.0069	0.2976	0.96486	(0.90206–1.03204)
	65–79	0.0472	0.0058	**<0.0001**	1.26602	(1.19578–1.34039)
	80+	0.0720	0.0251	**0.0041**	1.43326	(1.12113–1.83230)
Nasopharynx	all	−0.0080	0.0048	0.0965	0.96067	(0.91626–1.00722)
	35–49	0.0122	0.0072	0.0896	1.06278	(0.99063–1.14018)
	50–64	−0.0314	0.0070	**<0.0001**	0.85465	(0.79807–0.91523)
	65–79	−0.0009	0.0081	0.9102	0.99542	(0.91914–1.07804)
	80+	0.0113	0.0399	0.7777	1.05796	(0.71549–1.56435)
Hypopharynx	all	−0.0139	0.0053	**0.0089**	0.93301	(0.88577–0.98276)
	35–49	−0.0942	0.0122	**<0.0001**	0.62452	(0.55419–0.70378)
	50–64	−0.0376	0.0059	**<0.0001**	0.82852	(0.78184–0.87799)
	65–79	0.0192	0.0048	**<0.0001**	1.10082	(1.05068–1.15336)
	80+	0.0445	0.0170	**0.0087**	1.24938	(1.05799–1.47539)
Larynx	all	0.0030	0.0039	0.4304	1.01532	(0.97766–1.05443)
	35–49	−0.0367	0.0078	**<0.0001**	0.83220	(0.77129–0.89791)
	50–64	−0.0191	0.0038	**<0.0001**	0.90896	(0.87574–0.94344)
	65–79	0.0135	0.0050	**0.0067**	1.06972	(1.01881–1.12317)
	80+	0.0497	0.0174	**0.0042**	1.28233	(1.08162–1.52028)

StdErr = standard error; RR = relative risk; CI = confidence interval; * significant values (*p* < 0.05) in bold.

## Data Availability

The datasets used during the current study are available from the corresponding author upon reasonable request. Data source: Research Data Centre of the Federal Statistical Office and Statistical Offices of the Länder, [DRG statistics], survey year(s) [2005–2018], own calculations.

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
