# Peer review of "The Association between Patient’s Age and Head and Neck Cancer Treatment Decision—A Population-Based Diagnoses-Related Group-Based Nationwide Study in Germany"

_cancers, 2023, doi:10.3390/cancers15061780_

Round 1

Reviewer 1 Report

The authors report on 2 things: type of therapy over time and type of therapy based on age group.

Thepaper does not have merit scientifically to justify publication.

No introduction to discuss trends in treatment in the selected period (for example TORS). No discussion of the impact of HPV. 
Morevoer, no stratification according to stage of disease and no discussion of the impact of the modifications over time on overall survival.

This paper is more a statistical report than a research question and does not fit the criteria for Cancere

Reviewer 2 Report

I have no comments to introduction.

I have no coments to ethics - in this type of study an approval of ethical committee is not required.

I have no comments to patient selection section - it is clearly described.

I have no comments to statistical analysis.

Results are clearly reported. It would be extremely interesting to find the explanation, why neck dissection (3.42), radiotherapy (5.05) and chemotherapy (4.87) rates were much lower than surgery rates (8.70). [row 116] or why [row 228] the increase in systemic therapy treatment rates were seen in the older age cohorts ≥65, especially in salivary glands (I am not aware of any effective systemic treatment for salivary carcinoma except trastuzumab in HER2 positive adenocarcinoma, which is a small group). It simply couldn't cause such a difference - it is extremely interesting.

However, at the same time, authors clearly stated: [row 281] - causal connections, for example between cancer diagnosis and chosen therapy, were only traceable to a limited extent. Also, authors clearly stated [row 283] that data included only all discharged, fully hospitalized patients. Day-care or outpatient patients were not included. They accept, that older HNC patients might have a higher probability for inpatient treatment.

There is a typing error row 289 - sentence "the majority" is typed twice.

I have no comment of discussion and conclusions.

I have no comments to references - all appropriate.

Author Response

I have no comments to introduction.

I have no comments to ethics - in this type of study an approval of ethical committee is not required.

I have no comments to patient selection section - it is clearly described.

I have no comments to statistical analysis.

I have no comment of discussion and conclusions.

I have no comments to references - all appropriate.

Answer 2.0: Thanks!

Results are clearly reported. It would be extremely interesting to find the explanation, why neck dissection (3.42), radiotherapy (5.05) and chemotherapy (4.87) rates were much lower than surgery rates (8.70). [row 116] or why [row 228] the increase in systemic therapy treatment rates were seen in the older age cohorts ≥65, especially in salivary glands (I am not aware of any effective systemic treatment for salivary carcinoma except trastuzumab in HER2 positive adenocarcinoma, which is a small group). It simply couldn't cause such a difference - it is extremely interesting.

Answer 2.1: Yes. These differences are very interesting. Unfortunately, the methodology did not allow to analyze causal relations nor to identify specific drugs (like the mentioned trastuzumab and others). The coding did not allow a separation between classical chemotherapy and biological treatment. This will become better in the future, as more patients receive nowadays biologicals/antibodies. We added a sentence in the Discussion on page 13 to address this problem.

However, at the same time, authors clearly stated: [row 281] - causal connections, for example between cancer diagnosis and chosen therapy, were only traceable to a limited extent. Also, authors clearly stated [row 283] that data included only all discharged, fully hospitalized patients. Day-care or outpatient patients were not included. They accept, that older HNC patients might have a higher probability for inpatient treatment.

Answer 2.2: Yes. Outpatient treatment is a black box here. This question was also answered by reviewer #3, see answer 3.3. We added in the Discussion on page 13 starting in line 295: “Almost all cases of surgery for HNC are performed as inpatient cases. In contrast, since about 10 years there is an increasing tendency to perform radiotherapy, especially when performed as a single modality, as outpatient treatment. If the patients receive radiochemotherapy, most patients are still treated as inpatients in the weeks receiving chemotherapy. Numbers on the portion of outpatient radiotherapy treatment for HNC in Germany are not published. Nowadays, most radiation oncologists would recommend outpatient radiotherapy even in elderly HNC patients [28]. Day-care or outpatient patients were not included. Hence, the overall number of patients receiving radiotherapy for HNC in Germany is probably higher, especially in the most recent years”.

There is a typing error row 289 - sentence "the majority" is typed twice.

Answer 2.3: We corrected the sentence.

Reviewer 3 Report

Dear authors, it was very interesting to read the well-written manuscript. The growing subpopulation of elderly HNSCC patients is a major issue we face in daily clinical practice, therefore this is one of the hot topics in head and neck oncology.

Before this paper is ready for publication I would like to raise a few questions:

1. What is the rationale to choose "biopsy" as treatment option? This was very confusing to me. Biopsy is mentioned as treatment in table 1 and throughout the manuscript. Please explain and reconsider, if biopsy needs to be analyzed?

2. Colors in Figure 1 should be different. We do you use yellow for 35-50 and > 80 years?

3. In the limiatations of the study you decribe that "Day-care or outpatient patients were not included. Older HNC patients might have a higher probability for inpatient treatment. Elective medical services, for example a desired treatment by a chief physician, were not billed via the DRG system". Does this mean that the majority of outpatient performed radiation or chemoradiation therapy is not included in this study? Especially single modality treatment by radiotherapy is performed in an outpatient setting in Germany. Please explain and maybe clearly describe which treatments are probably missing due to the methodology of the study. Please mention an estimated percentage of missed treatments or cases in relation to newly diagnosed patients in the study period. 

Author Response

Dear authors, it was very interesting to read the well-written manuscript. The growing subpopulation of elderly HNSCC patients is a major issue we face in daily clinical practice, therefore this is one of the hot topics in head and neck oncology.

Before this paper is ready for publication I would like to raise a few questions:

3.1. What is the rationale to choose "biopsy" as treatment option? This was very confusing to me. Biopsy is mentioned as treatment in table 1 and throughout the manuscript. Please explain and reconsider, if biopsy needs to be analyzed?

Answer 3.1: Thanks for this question. We added an explanation in the Methods on page 3: “Biopsies as tumor biopsies taken during panendoscopy were also included, although taking biopsies is not a treatment in the strict sense. Panendoscopy including taking biopsies is a standard upstream diagnostic procedure in general anesthesia for treatment decision making.in Germany. Therefore, the procedure of taking these biopsies were also analyzed in this study.”

3.2. Colors in Figure 1 should be different. We do you use yellow for 35-50 and > 80 years?

Answer 3.2: We changed the colors. Now the four age cohorts can easily be differentiated.

3.3. In the limitations of the study you describe that "Day-care or outpatient patients were not included. Older HNC patients might have a higher probability for inpatient treatment. Elective medical services, for example a desired treatment by a chief physician, were not billed via the DRG system". Does this mean that the majority of outpatient performed radiation or chemoradiation therapy is not included in this study? Especially single modality treatment by radiotherapy is performed in an outpatient setting in Germany. Please explain and maybe clearly describe which treatments are probably missing due to the methodology of the study. Please mention an estimated percentage of missed treatments or cases in relation to newly diagnosed patients in the study period. 

Answer 3.3: This is an important issue. Thanks to address this! 1) Since about 10 years there is an increasing tendency to perform radiotherapy as a single modality as outpatient treatment. If the patients receive radiochemotherapy, most patients are still treated as inpatients in the weeks receiving chemotherapy. Numbers on the portion of outpatient radiotherapy treatment for head and neck cancer in Germany are not published. Nowadays, most radiation oncologists would recommend outpatient radiotherapy even in elderly head and neck cancer patients (Haehl E, Rühle A, Spohn S, Sprave T, Gkika E, Zamboglou C, Grosu AL, Nicolay NH. Patterns-of-Care Analysis for Radiotherapy of Elderly Head-and-Neck Cancer Patients: A Trinational Survey in Germany, Austria and Switzerland. Front Oncol. 2022 Jan 3;11:723716. doi: 10.3389/fonc.2021.723716. PMID: 35047384; PMCID: PMC8761738.). We added this information in the Discussion starting in line 295: “Almost all cases of surgery for HNC are performed as inpatient cases. In contrast, since about 10 years there is an increasing tendency to perform radiotherapy, especially when performed as a single modality, as outpatient treatment. If the patients receive radiochemotherapy, most patients are still treated as inpatients in the weeks receiving chemotherapy. Numbers on the portion of outpatient radiotherapy treatment for HNC in Germany are not published. Nowadays, most radiation oncologists would recommend outpatient radiotherapy even in elderly HNC patients [27]. Day-care or outpatient patients were not included. Hence, the overall number of patients receiving radiotherapy for HNC in Germany is probably higher, especially in the most recent years”. 2) We give a number now for the treatments outside the DRG systems. It is 10% (on page 13, line 306).

Reviewer 4 Report

The authors have conducted an extensive analysis of H@N treatment decisions using a high quality national cancer registry in Germany There is value here, as the data is inclusive of all cases in the population followed over a long time period. 

The premise of this study, as I understand it, is that older patients are underrepresented in clinical trials, which could be an issue given the growing population of older patients. The authors cite a few studies to support this, but in looking at these references I either did not see data to support this or the references were quite old or based on a single study or institution. The German Clinical Trials Register for example, likely has age information on head and neck trials in Germany. In the U.S. the clinicaltrials.gov registry has numerous head and neck trials listed with age information. 

There are many tables and it is somewhat difficult to get a sense of the information with over 150 statistical tests.  Most are significant  due to the large sample size. The negative binomial regression may not be the best way to analyze the data. It assumes a linear relationship over time, which is true for some of the trends but not for others, at least based on inspection of the figure. The windows-based joinpoint regression program might be worth considering. 

My sense of the data is that over this long time period, there are a number of factors not discussed that may be important to the interpretation. For example, oropharynx has become largely HPV-related, and so are the treatment trends for oropharynx impacted by this and in the expected direction?  

The discussion reviews age-related differences in cancer management but doesn't directly address the time trend analysis. SO while older patients are treated less aggressively, the time trend analysis indicates that over the years, there is an increase in treatment for older age groups. Perhaps the increase is small but real nevertheless and supposedly that is a welcome trend? Authors should discuss.  A main finding mentioned in the discussion is the decreasing rates of most cancers in younger patients. However the reasons for this are not discussed, or the implications.  

The figure needs a label for the Y-axis. I assume it is rates per 100,000. Also the color for the 35-<50 age group is the same to that for the >80 group. Suggest changing the color code for one of those groups to red. 

Author Response

The authors have conducted an extensive analysis of H@N treatment decisions using a high quality national cancer registry in Germany There is value here, as the data is inclusive of all cases in the population followed over a long time period.

The premise of this study, as I understand it, is that older patients are underrepresented in clinical trials, which could be an issue given the growing population of older patients. The authors cite a few studies to support this, but in looking at these references I either did not see data to support this or the references were quite old or based on a single study or institution. The German Clinical Trials Register for example, likely has age information on head and neck trials in Germany. In the U.S. the clinicaltrials.gov registry has numerous head and neck trials listed with age information. 

Answer 4.1: The study included patients from 2005 to 2018. These patients were treated according to the clinical guidelines of the years 2005 to 2018 (hopefully). These guidelines are based mainly based on the level I clinical trials of the previous years. All milestone trials for these years did not include elderly patients. We think that the situation has improved very much. We followed the advice of the reviewer #4 and checked clinicaltrials.gov (important German studies are also registered there): On 22-Feb-2023 the search with the terms “head and neck cancer”, “older”, and “elderly” eleven studies (https://www.clinicaltrials.gov/ct2/results?cond=head+and+neck+cancer&term=elderly+old&cntry=&state=&city=&dist=)! Only one (1) of these studies is a phase III trial!. Hence, we added in the Introduction on page 2, beginning with page 45: “Despite of this, clinicaltrials.gov only lists at the moment eleven trials focused on el-derly HNC patients. And only one of the studies is a phase III trial (https://www.clinicaltrials.gov/ct2/results?cond=head+and+neck+cancer&term=elderly+old&cntry=&state=&city=&dist=; access: 22-February-2023).”

There are many tables and it is somewhat difficult to get a sense of the information with over 150 statistical tests.  Most are significant  due to the large sample size. The negative binomial regression may not be the best way to analyze the data. It assumes a linear relationship over time, which is true for some of the trends but not for others, at least based on inspection of the figure. The windows-based joinpoint regression program might be worth considering. 

Answer 4.2: The large sample size was indeed an important strength of this study. This allowed meaningful statistical analyses. Thank you for your thoughtful remark and the suggestion concerning the joinpoint regression program. Peter Schlattmann, coauthor and statistician addressed this question: Looking at the paper by Kim et al (2000) joinpoint regression relies on normally or Poisson distributed errors (Kim HJ, Fay MP, Feuer EJ, Midthune DN. Permutation tests for joinpoint regression with applications to cancer rates. Stat Med 2000;19:335-51 (correction: 2001;20:655). When assuming a Poisson distribution the problem of overdispersion occurs. This is why we chose a negative binomial model which accommodates overdispersion. Furthermore, we had a look at the software. As far as we understand it besides age adjustment no further adjustment for other covariates is possible. In terms of linearity negative binomial regression is a generalized linear model. The linear predictor can be modified by e.g. including polynomial terms. Since we have overdispersed data and several other covariates besides age we decided to stick to the negative binomial regression model.

My sense of the data is that over this long time period, there are a number of factors not discussed that may be important to the interpretation. For example, oropharynx has become largely HPV-related, and so are the treatment trends for oropharynx impacted by this and in the expected direction?  

Answer 4.3: We added some sentences in the Discussion on page 12, beginning with line 285: “Tumor stage as well as the HPV status are not coded in the DRG system. The hospitals (not the DRG system) registers the HPV status for patients with oropharyngeal cancer since introduction of the 8th edition of the TNM classification in 2017. The present study included all cases until 2018. Hence, an influence of the HPV-status on decision making could only be marginal. As in many other countries, we see an increase of oropharyngeal cancer in the recent years. This increase is probably related to an increase of HPV+ oropharyngeal cancer (12]. This might explain why treatment rates for oro-pharyngeal cancer have increased in the recent years.”

The discussion reviews age-related differences in cancer management but doesn't directly address the time trend analysis. So while older patients are treated less aggressively, the time trend analysis indicates that over the years, there is an increase in treatment for older age groups. Perhaps the increase is small but real nevertheless and supposedly that is a welcome trend? Authors should discuss.  A main finding mentioned in the discussion is the decreasing rates of most cancers in younger patients. However the reasons for this are not discussed, or the implications.  

Answer 4.4: Thank you for this important comment as it addresses the main topic of the study.

Concerning the relative increase use of biopsies and surgery in older HNC patients, we added in the Discussion on page 11: “This increase in elderly patients was most pronounced for biopsies and surgeries. The increase for biopsies can easily explained by the increasing incidence rates of HNC in elderly patients, thus needing this standard diagnostic procedure. The surgery rates might have increased because screening tools were developed in the recent years al-lowing a better risk stratification and individual treatment deintensification decisions in elderly patients [15].”

Concerning the relative increase use of chemotherapy/biologicals in older HNC patients, we write now in the Discussion on page 12: “Nevertheless, the relative increase in chemotherapy/biologicals use over time was largest in patients ≥ 65 to <80 years. The data structure did not allow a differentiation between chemotherapeutics and biologicals. It is rather plausible that the increased use is related to an increasing use of biologicals than of the chemotherapeutics. Cetuximab was licensed in 2004. Cetuximab is especially used in elderly HNC patients with worse performance status who are not ineligible to receive a platinum-based chemotherapy [27].”

We address now also the decrease of treatment rates in younger HNC patients, also in the Discussion on page 11: “We can only speculate about the reasons. This might be signs of the worldwide reduced tobacco and alcohol consumption. This effect reaches now first the younger age cohorts [16].”

The figure needs a label for the Y-axis. I assume it is rates per 100,000. Also the color for the 35-<50 age group is the same to that for the >80 group. Suggest changing the color code for one of those groups to red. 

Answer 4.5: See also Answer 3.2 (reviewer #3). We changed the colors. And we added a Y-axis. Doing this, we recognized that the scaling in 1C was different form the other graphs. This was changed, too.

Round 2

Reviewer 4 Report

Authors have appropriately responded to comments. THe paper adds to this literature.